# Proton-controlled molecular ionic ferroelectrics

Yulong Huang [1] ✉, Jennifer L. Gottfried [2], Arpita Sarkar[1], Gengyi Zhang[3], Haiqing Lin [3] & Shenqiang Ren [1,4,5,6] ✉

Molecular ferroelectric materials consist of organic and inorganic ions held together by hydrogen bonds, electrostatic forces, and van der Waals interactions. However, ionically tailored multifunctionality in molecular ferroelectrics has been a missing component despite of their peculiar stimuli-responsive structure and building blocks. Here we report molecular ionic ferroelectrics exhibiting the coexistence of room-temperature ionic conductivity $(6.1 \times 10^{-5}$ S/cm) and ferroelectricity, which triggers the ionic-coupled ferroelectric properties. Such ionic ferroelectrics with the absorbed water molecules further present the controlled tunability in polarization from 0.68 to 1.39 μC/cm², thermal conductivity by 13% and electrical resistivity by 86% due to the proton transfer in an ionic lattice under external stimuli. These findings enlighten the development of molecular ionic ferroelectrics towards multifunctionality.

Molecular ferroelectrics[1–3] are a class of materials that exhibit ferroelectric properties, promising for data storage, sensors, actuators, and electro-optics[4–8]. Molecular ferroelectrics are typically composed of ionic building blocks[9], enabling a possibility of the flow of ions[10,11]. However, the displacement of charged ions tends to impede the electron flow in electronic-insulating ferroelectric lattice. An interest arises in materials science if molecular ferroelectrics could be simultaneously ionic conductors that enable the flow of charged ions in the lattice, deemed as molecular ionic ferroelectrics (Fig. 1a). To achieve such integration, we surmised the following design parameters: (1) molecular ferroelectrics exhibit high-temperature ionic conductivity and spontaneous polarization (high Curie temperature); and (2) ionic-polarization dual nature of molecular ferroelectrics with stimuli-responsive behavior.

Here we select imidazolium perchlorate (ImClO₄) for a prototypical exampled candidate[12–14], a solid-state molecular ionic ferroelectric crystal, to show ionic conductivity and ferroelectricity, combining properties of conventional ionic conductors[15–22] and ferroelectrics[23–28] (Fig. 1a). Ionic-ferroelectric ImClO₄ composes of imidazolium cation ($C_3H_4N_2^+$) and perchlorate anion ($ClO_4^-$). Those cation and anion arrays are alternatively arranged and locate in (101) planes (Fig. 1b). Given the fact that ImClO₄ is crystallized in an aqueous solution[12,14], the absorbed water molecules coordinate by hydrogen bonds and function as proton reservoirs in the lattice. The mobile protons can diffuse in ImClO₄ lattice via the Grotthuss mechanism[29,30]. The spontaneous polarization of ImClO₄ results from the dipoles induced by displacements of $C_3H_4N_2^+$ cation and $ClO_4^-$ anion in the crystal lattice, occurring below its Curie temperature of 373 K[12,14].

For a non-spontaneously polarized configuration above Curie temperature, Landau free energy maintains a minimum at the zero polarization. For a spontaneous polarized configuration of ferroelectrics (e.g., ImClO₄) at room temperature, the equilibrium states are stabilized by a spontaneous polarization that is corresponding to one of the degenerate energy minima in the double-well Landau free energy[31–33] (Fig. 1c). Once external biases, such as ions or electric field, are applied on a spontaneously polarized ferroelectric system, the

[1]Department of Mechanical and Aerospace Engineering, University at Buffalo, The State University of New York, Buffalo, NY 14260, USA. [2]Weapons Sciences, US Army Combat Capabilities Development Command-Army Research Laboratory, Aberdeen Proving Ground, Aberdeen, MD 21005, USA. [3]Department of Chemical and Biological Engineering, University at Buffalo, The State University of New York, Buffalo, NY 14260, USA. [4]Department of Chemistry, University at Buffalo, The State University of New York, Buffalo, NY 14260, USA. [5]Research and Education in Energy, Environment and Water (RENEW) Institute, University at Buffalo, The State University of New York, Buffalo, NY 14260, USA. [6]Department of Materials Science and Engineering, University of Maryland, College Park, MD 20742, USA. ✉e-mail: yhuang59@buffalo.edu; sren@umd.edu

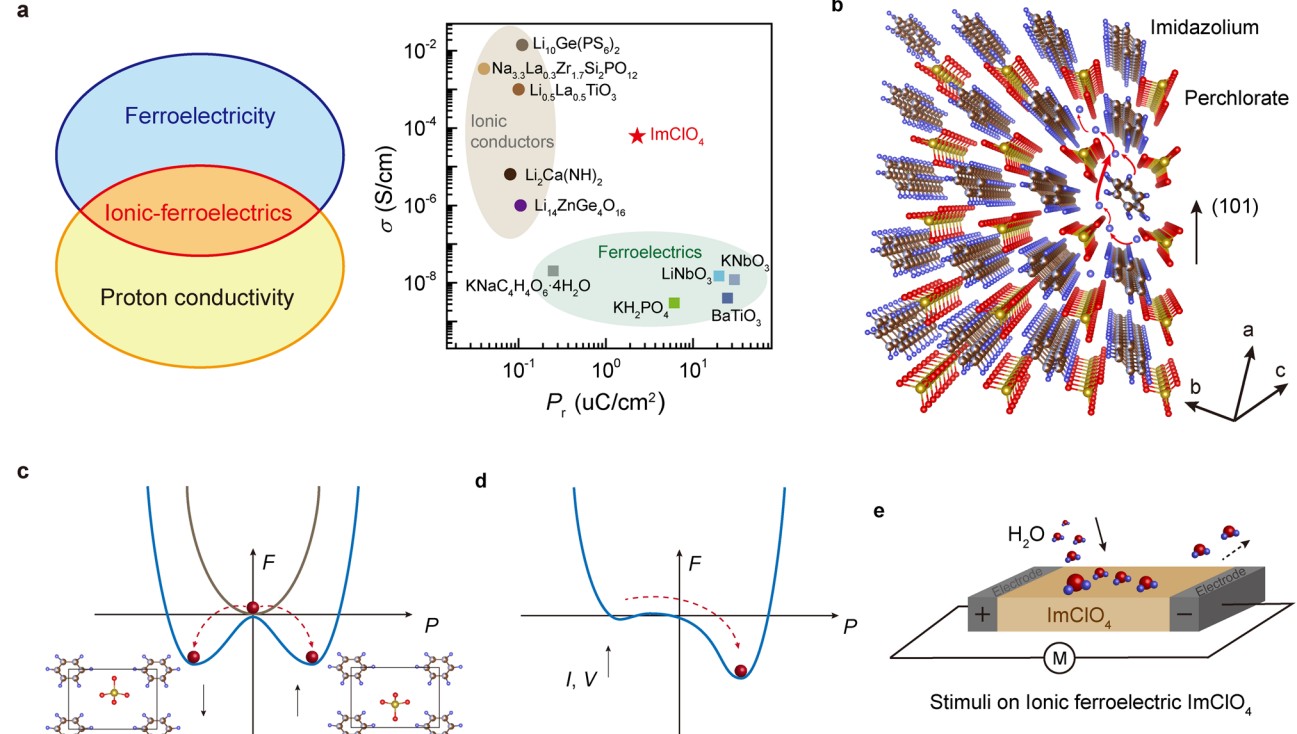

**Fig. 1 | Molecular ionic-ferroelectrics. a** Schematic illustration of ionic-ferroelectrics that show the coexistence of ferroelectricity and proton conductivity. $ImClO_4$, a molecular ionic-ferroelectric, shows its conductivity and remanent polarization among conventional ionic conductors and ferroelectrics. **b** Crystal structure of $ImClO_4$ shows the alternative stacking of imidazolium cations and perchlorate anions along (101) plane. The possible pathways of proton transfer are perpendicular to (101) plane, through hydrogen bonds between $ClO_4^-$ and imidazole molecules (red arrows). All atoms C (brown), H (blue), O (red), N (light blue), and Cl (yellow) are distinguished by their colors. **c** Landau free energy ($F$) versus polarization ($P$) curves show the spontaneously non-polarized and polarized states. **d** External biases (current/or voltage $V$) effect on Landau free energy versus polarization curve. **e** Schematic of an ionic-ferroelectric device based on $ImClO_4$ Crystal indicates proton effect. The electrical bias direction is perpendicular to (101) direction to reveal a coexistence of ferroelectricity and proton conductivity in ionic-ferroelectrics.

Landau free energy versus polarization landscape becomes asymmetric that the energy minima locate in one well depending on the bias. Therefore, external biases could tailor the relationship of Landau free energy and polarization in ionic ferroelectrics (Fig. 1d), where molecular ionic ferroelectric $ImClO_4$ provides a platform to study the coexistence of ionic/proton conductivity and ferroelectricity and their responses under external stimuli. As illustrated in Fig. 1e, external biases are applied perpendicular to (101) plane of $ImClO_4$ crystal, while the proton effects could be controlled by humidity.

## Results

The peculiar arrangement of ionic arrays in $ImClO_4$ presents the pattern of strips as revealed by scanning electron microscopic image (Fig. 2a). Molecular $ImClO_4$ crystals preferentially grow along (101) plane, presenting a set of X-ray diffraction (XRD) peaks that equally distributed in diffraction angles (Fig. 2b). High-quality $ImClO_4$ crystal presents a regular shape and transparency as shown in the inserted optical image. As an ionic crystal, $ImClO_4$ lattice vibrates in the modes contributed from imidazolium and perchlorate ions. The (symmetric and antisymmetric) stretching and (in-plane and out-of-plane) deformation modes[34] of anionic $ClO_4^-$ are clearly presented by Raman peaks (Fig. 2c). Electrochemical impedance spectrum[35] confirms ionic/proton conductance in $ImClO_4$ crystal. The Nyquist plot of the impedance data shows a linear relationship between the real (Z′) and imaginary (−Z″) parts with the slope of approximately 3 at the low-frequency region (Fig. 2d). By extrapolation of low-frequency straight line, the x-intercept is used to calculate ionic conductivity of $6.1 \times 10^{-5}$ S/cm. The plot develops into a semicircle contributed to ionic/proton conductivity in $ImClO_4$.

The classic P-E hysteresis loops expand with the range of applied electric field. The magnitude of electric field bias influences the polarization of $ImClO_4$ crystal, as shown in the plots of polarization-electric field (P-E) loops (Figs. 3a and S1)[8]. Differential scanning calorimetry (DSC) confirmed that $ImClO_4$ undergoes an endothermic phase transition from ferroelectric to paraelectric at around 373 K by heating, and an exothermic transition at around 365 K by cooling (Figs. 3b and S2). The sharp transition peaks around 212 and 246 K correspond to structural transition[36] during cooling and heating. The ability of heat conductance in $ImClO_4$ increases with temperature and saturates near the ferroelectric-paraelectric transition, indicating the electron-phonon coupling effect[37]. A reduced thermal conductivity is observed when $ImClO_4$ is cooled from 400 K (above its Curie temperature), implying ferroelectric ordering induced polarization-controlled thermal conductivity[38] (Fig. 3c). The proton/phonon contribution from polarized cations and anions could be considerable, as electron/proton-phonon coupling influences heat conductance[37–39]. Such entanglement between ferroelectric order and heat conductance is reliably observed in both powder and thin-film forms of molecular ionic ferroelectric materials (e.g., guanidinium perchlorate, Fig. S3). The transfer of proton ion influences the dipoles in an ionic ferroelectric[10,40,41], reflecting on relative permittivity as heating from a low temperature. A large temperature range of permittivity enhancement appears at around 300 K during heating, while a nearly constant permittivity remains during the cooling (Fig. 3d). Such anomaly in permittivity could be ascribed to excess mobile proton ions released from the framework during heating. Meanwhile, the amount of proton ions is reduced after heating up to 380 K, and the anomaly disappears when cooling down. The relative permittivity reaches a maximum at

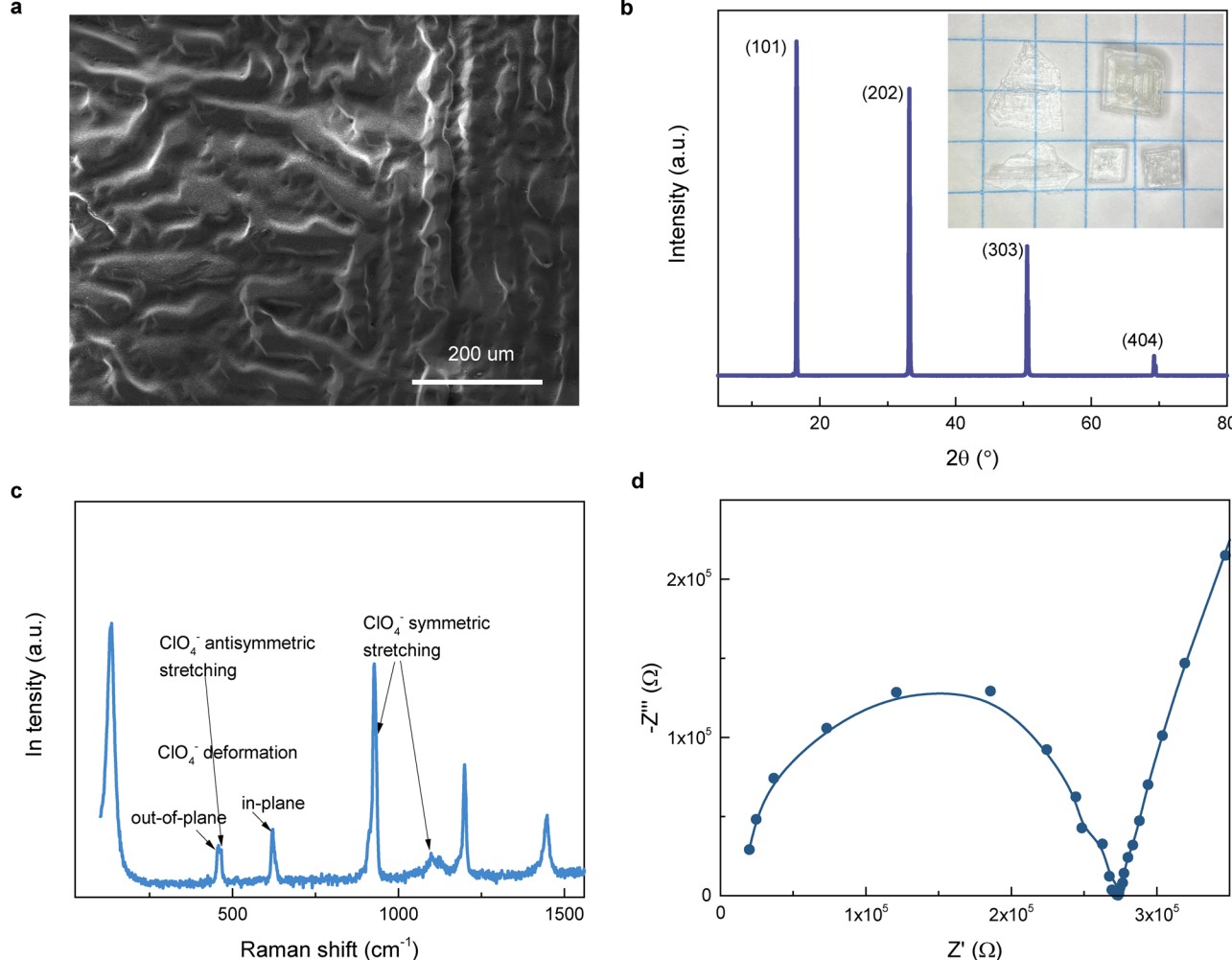

**Fig. 2 | Morphological, structural, and ionic conductance of ImClO₄ crystals.** **a** Scanning electron microscopic image indicates the surface morphology. **b** X-ray diffraction pattern of a flat ImClO₄ crystal shows the diffraction peaks of (101) plane. Optical images of ImClO₄ crystals show their transparency and shapes on a grid paper. One grid presents 5 mm. **c** Raman spectrum of ImClO₄ clearly indicates the molecular vibrations of imidazole and ClO₄. **d** Z′ vs Z″ plot of the impedance of ImClO₄ indicates ionic conductance.

372 K by cooling and 378 K by heating below its decomposition temperature (Fig. S4), while its temperature-dependent behavior is typical for a ferroelectric phase transition[13] (Fig. 3d). The permittivity anomaly near 300 K is repeatable and dependent on magnetic field, further suggesting the proton mechanism which relates to its cooling history (Figs. S5 and S6).

Ionic ferroelectric ImClO₄ becomes an appropriate platform to realize the control of proton ion for electron-phonon entangled process. Water molecules in ferroelectric lattice provide proton reservoir[29], while humidity dynamically determines the adsorption and desorption of water molecules of ImClO₄ to control the proton-dependent process. In ImClO₄, remanent polarization increases with the humidity and saturates at 50% relative humidity (Fig. 4a). Such humidity control enables the trapped water molecules to facilitate proton transfer in ferroelectric lattice. Ferroelectric performance exhibits humidity-control behavior, on which its remanent polarization recovers to the initial level when relative humidity decreases (Fig. S7). Time-resolved electrical resistivity of ImClO₄ in response to humidity indicates an effective role of proton transfer via the Grotthuss mechanism in the lattice. Figure 4b shows time-resolved electrical resistivity measurement on an ImClO₄ device at constant bias current (0.1 µA) and repeated cycles of controlled humidity (17 and 70%). Once ImClO₄ is periodically switched between a low humidity and a high humidity states, its resistivity decreases by about 86%.

Proton facilitates the change of electrical resistivity, indicating a modified electronic structure favored for a reduced resistivity because of proton-related coordination in lattice. Similarly, thermal conductivity shows a significant decrease when humidity increases from 17% RH to 70% RH (Fig. 4c). The thermal conductivity of molecular ferroelectrics is related to its polarization that is tunable by protons (Fig. 4a). The thermal conductance and polarization are intercorrelated to realize the proton-control of thermal conductivity. By applying electrical and thermal excitation, bidirectional thermal conductivity switching has been observed in antiferroelectric lead zirconate[39]. Compared to electric-field control[39,42], non-contact proton control possesses much convenience in the field of thermal circuit and thermal management. The measured thermal conductivity presents a high stability controlled by switching humidity. The XRD patterns confirm that no structural changes occur after humidification (Fig. 4d), while Raman spectra show no shifts of main vibrational modes of ImClO₄ under different relative humidity (Fig. S8). The PUND (positive up negative down) measurement and dielectric constant under different humidity confirm the ferroelectricity (Fig. S9) as well as the enhanced dielectric property (Fig. S10). Time-resolved infrared emission of ImClO₄ from millisecond-timescale combustion reactions following pulsed laser excitation indicates more thermally sensitive particles are present under a high humidity (the inset in Fig. 4d), suggesting the humidity effect on its thermal conductivity.

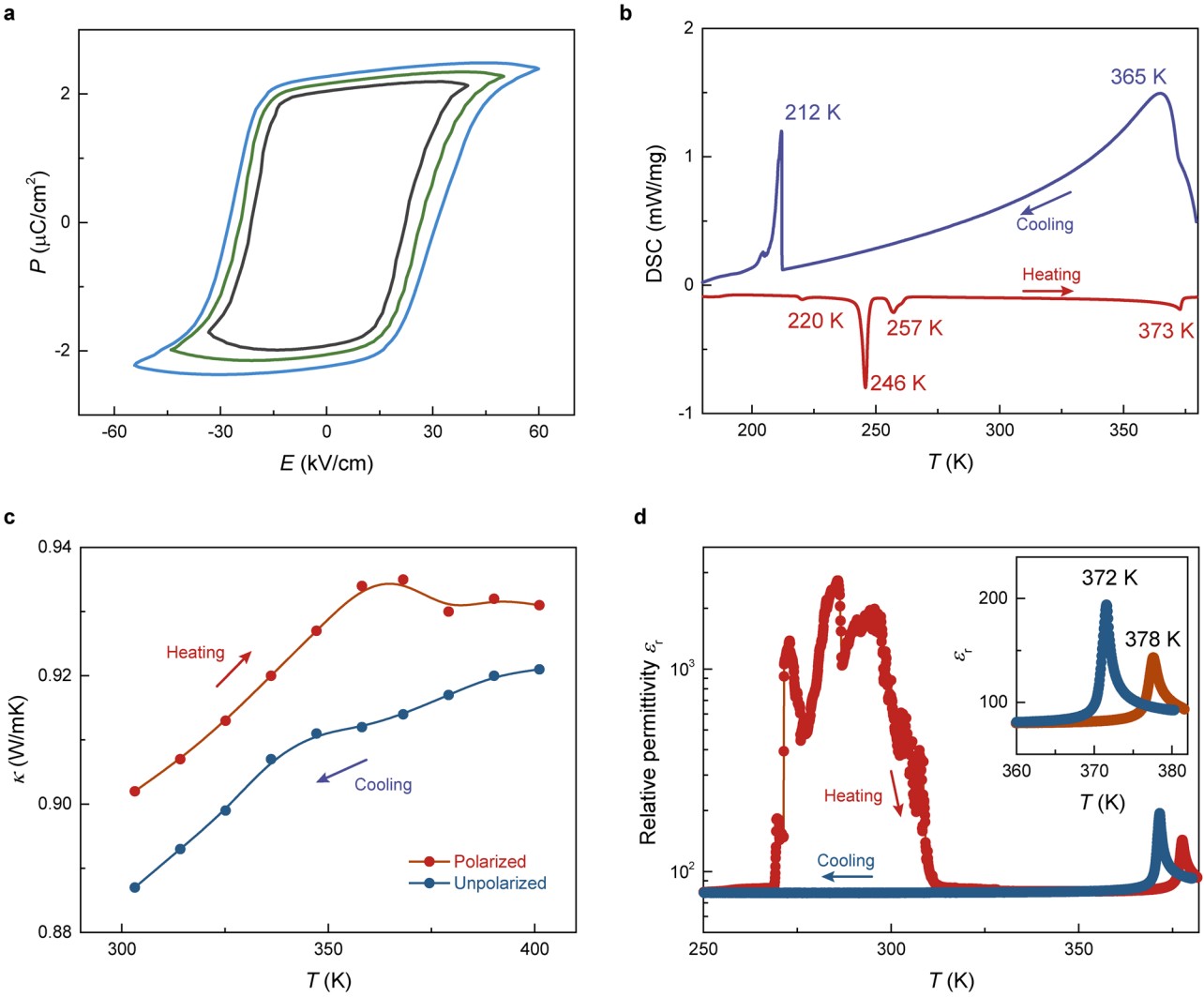

**Fig. 3 | Ferroelectricity, thermal, and electrical characterizations of ImClO$_4$ crystals. a** Polarization-electric field (*P-E*) hysteresis loops measured at different electric fields which are plotted in colors of black, green, and blue. **b** Differential scanning calorimetry (DSC) traces measured in the cooling and heating runs show the high-temperature ferroelectric phase transition and low-temperature structural transition. **c** Temperature-dependent thermal conductivity increases and then saturates near the ferroelectric ordering temperature. **d** Temperature-dependent relative permittivity indicates ferroelectric transition in heating and cooling measurements, while anomalous range near 300 K only appears in heating procedure. The enlarged inset clearly shows the permittivity peak due to the occur of ferroelectric transition.

Furthermore, the laser-induced air shock from energetic materials (LASEM) measurements show comparable shock and estimated detonation velocities before and after humidification, suggesting no structural change (Fig. S11 and Table S1). Fourier-transform infrared spectroscopy reveals the adsorption enhancement in ImClO$_4$ by increasing humidity from 21 to 70% (Figs. 4e and S12). The adsorption peaks at 3250 and 3155 cm$^{-1}$ are enhanced by 4.0% and 3.7%, respectively, implying the increased N−H stretching modes. This enhanced effect of N−H hydrogen bond may be ascribed to the increased proton transfer due to high humidity. Stronger H and O intensities from laser-induced plasma emission spectra are observed in the humidified ImClO$_4$, further evidencing the involvement of water molecules in the lattice after humidification while maintaining its structural integrity (Fig. 4f).

For a spontaneous polarized ferroelectric materials ImClO$_4$, the equilibrium state is corresponding to one of the degenerate energy minima in the double-well free energy (Fig. 1c). Once external biases (current or electric field) are applied onto ImClO$_4$, the free energy of system versus polarization landscape becomes asymmetric, promoting free charge carriers to move in the lattice. As the voltage bias

applied on ImClO$_4$ ferroelectric, the resistivity decreases from 284 to 122 MΩ·cm (Fig. 5a). Further increasing voltage continues to decrease resistivity and results in a low resistivity of about 35 MΩ·cm under 7.1 kV/cm. The reduced electrical resistivity indicates that proton in ImClO$_4$ lattice redistribute among ionic crystal lattice under electrical bias for a modified electronic structure with a promoted conductivity. When proton ions are relocated at a higher electrical bias, resistivity tends to maintain at a relatively low value compared to its initial state (Figs. 5a and S13). Due to the relocated proton, polarized charge is enhanced, so that the *P-E* loops are expanding in remanent polarization as ImClO$_4$ shows a decreasing resistivity (Figs. 5b and S14a). The slightly enhanced remanent polarization is observed under the biases of both 0.1 and 1 μA (Fig. S14b). With an increased current bias of 10 μA, ImClO$_4$ maintains a classic *P-E* loop in a much low resistivity (Fig. S14c). The expansion of *P-E* loop is reversible when ImClO$_4$ recovers into the initial high resistive state (Fig. S14d). The Raman spectra measured at different resistive states show no observable shifts of molecular vibration frequencies (Figs. S15, 16), excluding a change of crystal structure. Inspired by electrical bias tuning resistivity, ImClO$_4$ crystal is applied with a current bias switching between 0.1 and 1 μA to

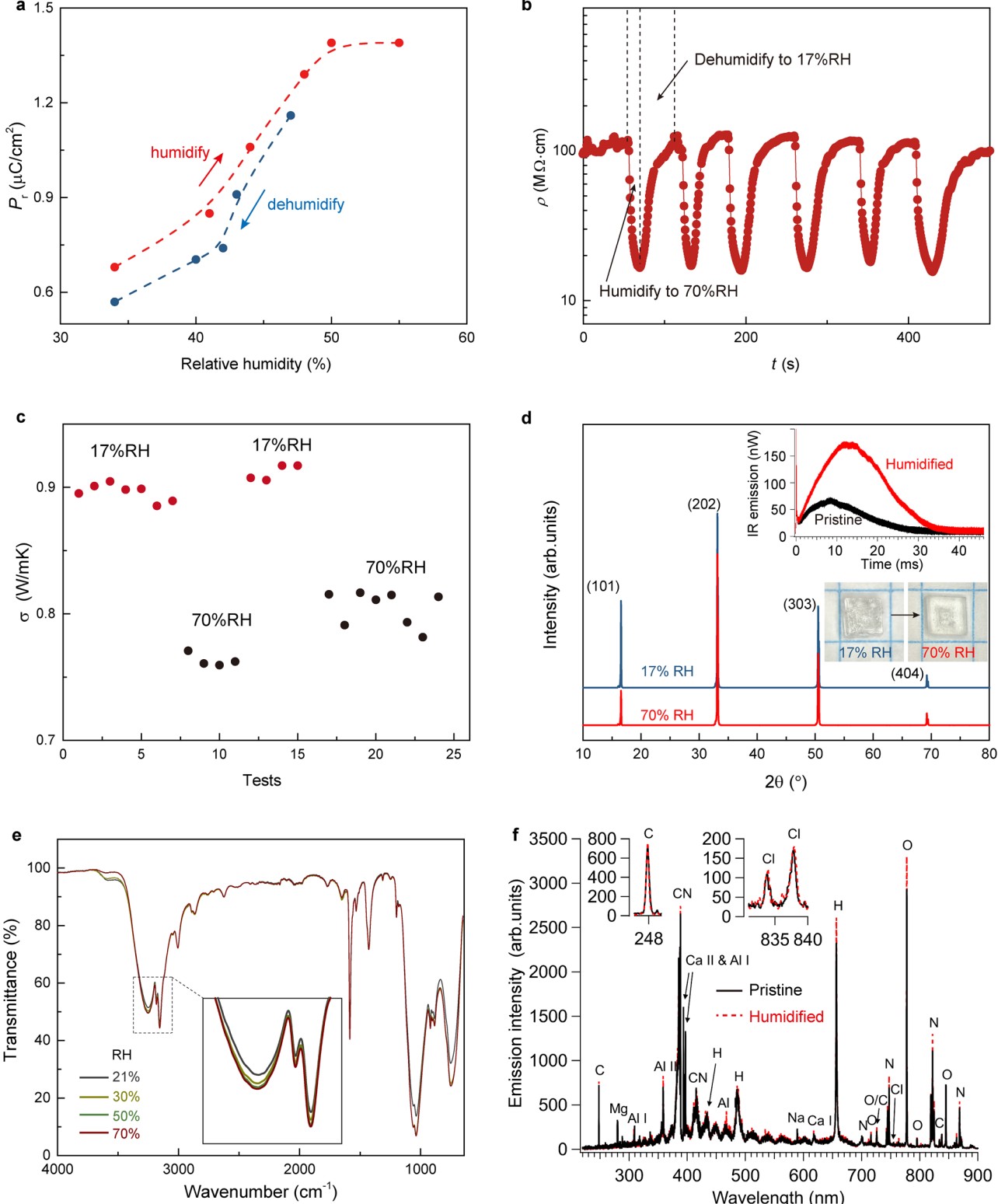

**Fig. 4 | Proton controlling on polarization and conductance in ImClO₄.**
**a** Humidity-dependent remanent polarization reveals an enhancement of ferro-electric performance. **b** A dynamic tuning of resistivity of ImClO₄ crystal by peri-odically switching humidity between 70 and 20%. **c** The thermal conductivity is also tuned by humidity with a high reproducivity. ImClO₄ crystal possesses a relatively high thermal conductivity under a high humidity. **d** ImClO₄ maintains the structure after humidifying. Time-resolved infrared emission indicates enhanced combustion from the more thermally sensitive particles. Optical images of ImClO₄ crystals show their morphologies before and after humidification. One grid pre-sents 5 mm. **e** Humidity increases the absorption intensity in ImClO₄. **f** Laser-induced plasma emission spectra of humidified ImClO₄ results in stronger H and O intensities, confirming the contribution of water molecules. The inserted figures show emission intensities of C and Cl remain unchanged in pristine and humidified ImClO₄ crystals.

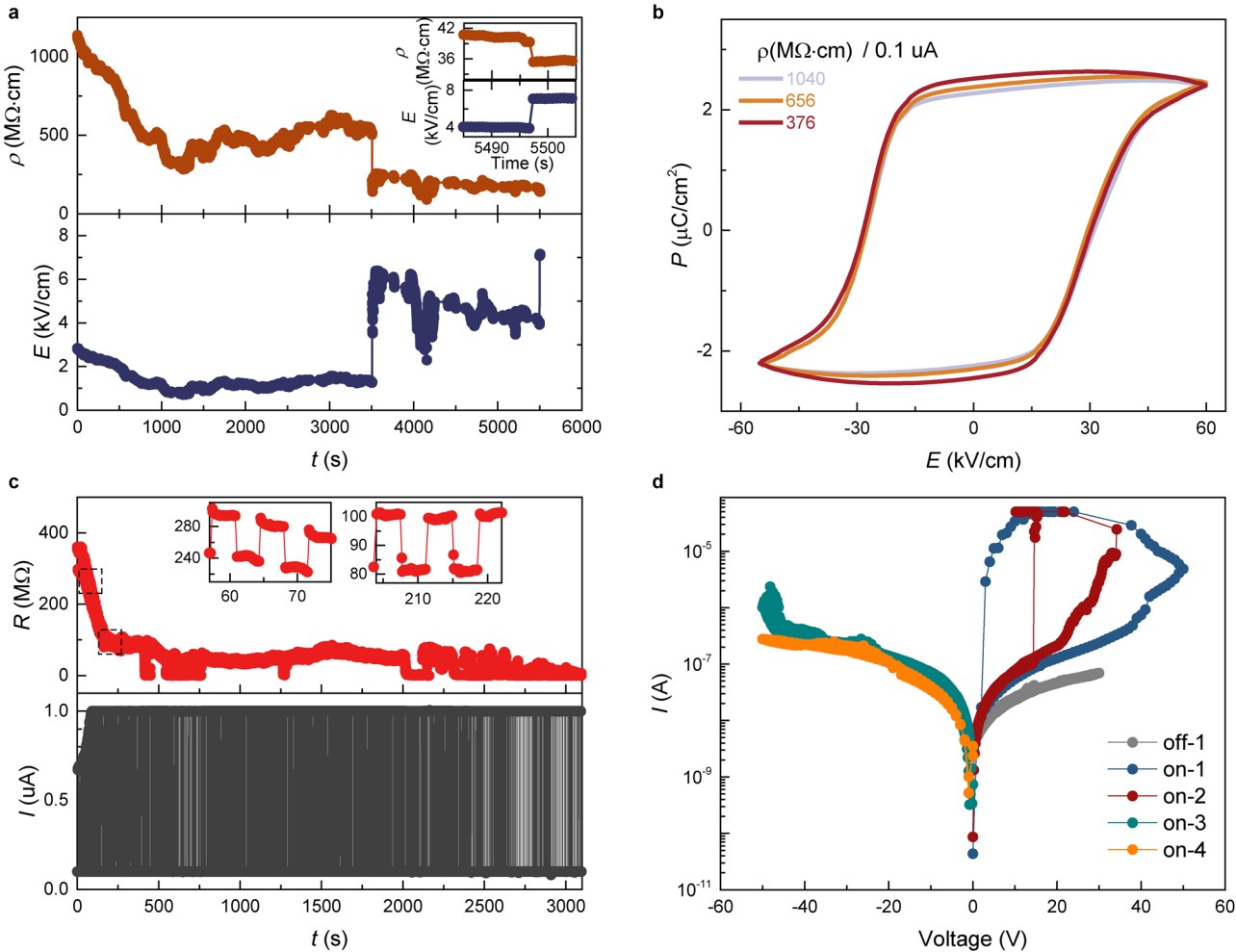

**Fig. 5 | Polarization alignment and memory effect. a** Tunable electric resistivity shows its continuous decrease by current. **b** The *P-E* loops are enhanced by relatively decreasing resistivity. **c** Periodical switching of current trains ImClO₄ into a low-resistance state. The resistance difference is revealed at 0.1 and 1 μA. After a procedure of decrease in resistance, the resistance change becomes stable as enlarged in the insets. **d** The voltage-current curve before and after current training. The memory effect with a sharp current jump only occurs along the direction of current training.

dynamically control its electrical resistivity (Fig. 5c). The resistance decreases and could be switched between high and low-resistance states. The resistance of both high and low-resistance states tends to be stable for over thousands of switching cycles. After electric bias, current versus voltage (*I–V*) curves are measured and a sharp jump is found that is absent in the initial *I–V* curve of ImClO₄ (Fig. 5d). Only positive bias results in a sharp and repeatable jump in *I–V* curves, which corresponds to the direction of switching electric bias in Fig. 5c. This electric-resistance switching behavior implies molecular ionic ImClO₄ ferroelectric material for a promising memory application.

## Discussion

The tunability of thermal conductivity, polarization, and resistivity is ascribed to proton transfer in ionic ferroelectric ImClO₄. The displacement of imidazolium cations and perchlorate anions results in the spontaneous polarization, meanwhile their ionic functional groups of nitrogen and oxygen provide conjunction sites for hydrogen bonds. The absorbed water molecules serve as the proton reservoir, allowing the Grotthuss diffusion among those conjunction sites via hydrogen bonds. On one hand, proton as an ion can adjust the polarized charge for a tunable ferroelectric performance; on the other hand, it influences electronic structure via hydrogen bonds to reduce electrical resistivity. Proton transfer is largely allowed through the breaking and formation of N–H and O–H bonds in the alternatively arranged

imidazolium and perchlorate ionic arrays. Besides, water molecules in ImClO₄ lattice provide proton-conducting channels. Proton transfer occurs along the stacking direction and passes through (101) planes of ImClO₄ crystal, revealing a coexistence of ferroelectricity and proton conductivity in molecular ionic-ferroelectrics.

In summary, molecular ionic ferroelectrics are shown as multifunctional materials that possess ferroelectricity and ionic conductivity in a single phase. We introduce imidazolium perchlorate (ImClO₄) as molecular ionic ferroelectrics with the coexistence of ferroelectricity and ionic conductivity, which simultaneously shows ionically controlled ferroelectric behavior. As an ionic ferroelectric with the absorbed water molecules in lattice, ImClO₄ exhibits its flexible tunability in remanent polarization, thermal conductivity, and electrical resistivity by humidity and external biases. This study provides the understanding of ion/polarization/thermal/resistive properties of molecular ferroelectrics and opens an effective route to explore more bifunctional materials based on molecular ferroelectrics.

## Methods

### Preparation of molecular ferroelectrics

The single crystals of molecular ferroelectric ImClO₄ were synthesized via a slow evaporation of an aqueous solution of imidazolium chloride and perchloric acid with 1:1 molar ratio. The obtained ImClO₄ crystals were dissolved into a saturated solution for recrystallization.

## Electrical and dielectric measurements

Polarization-electric field (*P-E*) hysteresis loops were measured on a Precision LC Ferroelectric Tester with the addition of a high voltage interface and a Trek 609B high voltage amplifier (Radiant Technologies Inc., USA). Humidity-dependent *P-E* loops were conducted by putting ImClO4 samples into a container where humidity was controlled by a humidifier. Electrical biases tuning and current-voltage curves were conducted on a Keithley 2450 SourceMeter. Once different resistive states were obtained, ImClO$_4$ sample was switched into another circuit for ferroelectricity measurements. The relative permittivity was measured on a precision impedance analyzer (Agilent 4294 A). A high-voltage dielectric probe (Radiant Technologies, Inc.) was mounted onto a physics properties measurement system (PPMS) for temperature and magnetic field control.

## Electrochemical impedance spectra

Squidstat Plus (Admiral Instruments) was used to measure electrochemical impedance spectra of ImClO$_4$ crystal from 1 MHz to 0.1 Hz. The ImClO$_4$ crystal was mounted in a Swagelok battery test cell and connected at the surface by two carbon electrodes.

## Fourier-transform infrared spectra

Transmittance spectra of ImClO$_4$ crystals were collected on an Agilent Cary 630 FTIR spectrometer. Humidity was controlled by a humidifier. Moisture was kept to blow onto sample stage until a certain humidity was reached.

## Structural and morphologic characterizations

A Rigaku Ultima IV (40 kV, 44 mA) was used to characterize the crystal structures of ImClO$_4$ crystals by X-ray diffraction. The surface morphology was measured on Carl Zeiss AURIGA (200 kV) Field Emission Scanning Electron Microscope (FESEM). The element analysis was determined by Oxford energy-dispersive X-ray spectrometer (EDS).

## Spectroscopy measurements

Raman spectra of ImClO$_4$ crystals were collected on Renishaw inVia Raman Microscope. The excitation wavelength is 785 nm.

## Thermal conductivity measurements

The thermal conductivity was measured on ferroelectric powder and film samples by using a Hot Disk TPS 2200 instrument (Hot Disk AB, Sweden). Ferroelectric films were prepared by spin-coating saturated aqueous solutions on FTO substrates. Since the ferroelectric sample is spontaneously polarized at room temperature, the thermal conductivity measured during heating was starting from a polarized state. When cooling from high temperature, the sample was in a unpolarized state. The humidity-dependent thermal conductivity measurements were conducted at room temperature.

## Differential scanning calorimetry

Heat flow of ImClO$_4$ from 176 to 381 K was detected using Differential Scanning Calorimetry (DSC, Q2000, TA Instruments, DE).

## LASEM measurements

The LASEM-420 system[43] was used to measure laser-induced shock velocities, plasma emission spectra, and time-resolved infrared emission. Before measurement, humidified crystal has been placed in an atmosphere with a relative humidity of 54% for half an hour. Another pristine crystal was measured as a reference. Both samples were ablated, atomized, ionized, and excited by a 6 ns pulsed laser (1064 nm) with the power of 850 mJ or 180 J/cm$^2$. The laser-induced shock velocities were collected using high-speed Schlieren imaging (420 kfps; 369 ns shutter) and were used to estimate the detonation velocity of the material. A high-resolution echelle spectrometer equipped with an intensified charge coupled device detector (Catalina Scientific SE200 with Apogee detector; gate delay = 1.5 μs, gate width = 10 μs, 200–1000 nm, $\lambda/\Delta\lambda = 2700$) was used to collect the emission spectra. An IR-sensitive photoreceiver (New Focus model 2053; 900–1700 nm) was used to monitor the integrated emission from the combusting particles on the millisecond timescale. Data from 20 laser shots were collected from each sample.

## Data availability

All relevant experimental data are presented in the paper and the Supplementary Information. Additional data related to this paper can be provided by the corresponding author upon reasonable request.

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

## Acknowledgements
The U.S. Department of Energy, Office of Basic Energy Sciences, Division of Materials Sciences and Engineering supports S.R. under Award DE-SC0023433.

## Author contributions
Y.L.H. and S.Q.R. conceived the idea and designed the study. Y.L.H. conducted all sample syntheses, structural characterizations and dielectric, electric measurements. A.S. provided help on thermal conductivity measurements. J.L.G. conducted the LASEM measurements. G.Y.Z. and H.Q.L. contributed to DSC measurement. Y.L.H. and S.Q.R. wrote the manuscript with comments and inputs from all authors.

## Competing interests
The authors declare no competing interests.
