## [Peer Review File · Nature Communications]

Proton-controlled Molecular Ionic FerroelectricsREVIEWER COMMENTS

Reviewer #1 (Remarks to the Author):

The authors investigated molecular ionic ferroelectrics exhibiting the coexistence of room-temperature ionic conductivity (6.1×10^{-5} S/cm) and ferroelectricity, which triggers the ionic-coupled ferroelectric properties. Such ionic ferroelectrics with aqua-ligand molecules further present the controlled tunability in polarization from 0.68 to 1.39 C/cm², thermal conductivity by 13 % and electrical resistivity by 86 % due to the proton transfer in an ionic lattice under external stimuli. In addition, the current version of this paper does not provide the insight result for the molecular ferroelectric ACTA PHYSICA SINICA 69 (21)s to guarantee the publication in Nature Communications. The following issue should be addressed before the publication.

1) The same material ImClO₄ were reported in the previous paper of these authors. The reference No. 30 is proton conductivity in the paper of Nat. Commun. 12, 255 4602 (2021). However, the same ImClO₄ is also ionic ferroelectrics.

2) What is the physics of ionic ferroelectrics? The author should discuss more about the physics.

3) Figures 1 c and d are not London free energy. It could be Landau free energy or total free energy. Normally, the free energy can be represented by F, and can not use E. In addition, the following related references should be cited to describe the free energy vs polarization in molecular ferroelectrics. Advanced Functional Materials 31 (38), 2104393, ACTA PHYSICA SINICA 69 (21).

4) In Figure 3a, the three polarization loops are under different electric field. However, the saturation polarizations are not equal for different electric fields.

5) Could the author explain the difference of phase transition during cooling and heating?

Reviewer #2 (Remarks to the Author):

Huang et al. reported the molecular ionic ferroelectric ImClO₄, which exhibits coexistence of ionic conductivity and ferroelectricity. Interestingly, the coexistence of ionic conductivity and ferroelectricity can trigger the ionic-coupled ferroelectric properties, i.e., ionic controlled ferroelectric behavior. Because ferroelectric-ion coupling is of great significance in the development of the devices with multi-well polarized states and neuromorphic characteristics, the starting point of this work is interesting and worthy of affirmation. However, there are still many problems to be clarified or solved in the manuscript before publication of the work. The main problems are as follows:

1) The polarization value of the P-E curve in the manuscript is much lower than that of the same compound reported in the literature, and even lower than that of the same compound reported by the authors themselves (Sci. Adv. 2017, 3, e1701008), why?

2) The change of P-E curve with humidity shows that with the increase of humidity, P-E curve appears certain leakage. Is it due to other contributions, rather than an actual increase in polarization?

3) Under humidity control, does water adsorb on the surface or enter the lattice? How to exclude the effect of water on simple surface adsorption of the sample?

4) Does ImClO₄ sample itself absorb moisture? Will the surface of the sample be damaged during this process?

5) In what way is the thermal conductivity measured? As we all know, it is difficult to obtain accurate thermal conductivity data, please provide details of this part, especially the thermal conductivity test before and after polarization and at different humidity.

6) In Figure 3, the author attributes the dielectric anomaly in a very wide temperature region near 300 K to the mobile proton released from the framework. However, this anomaly also exists in the

heating stage of DSC, which is attributed to the structural phase transition. Why?

7) In Figure 5b, the P-E loops are enhanced by relatively decreasing resistivity. However, the change is too small to be convincing.

8) For Figures S5 and S6, the authors claim that "The repeatable permittivity peak at ferroelectric ordering temperature indicates high crystalline quality of as-grown ImClO₄, further evidencing the proton mechanism for the anomaly in permittivity which relates to its cooling history and magnetic field dependence (Figs.S5 and S6)." However, the authors have been discussing dielectric anomalies around 300 K. Because the phase transition temperature is about 373 K, we have not yet seen repeatable peaks in permittivity at ferroelectric ordered temperatures. How to explain the variation of dielectric constant under cooling history and magnetic field dependence by proton mechanism? The author needs to provide a detailed explanation.

Reviewer #3 (Remarks to the Author):

This work reports the ionic tuning of molecular ferroelectricity, thermal transfer and electronic memory, which is a new exciting frontier for molecular ferroelectrics. The proton transfer in molecular crystals (ferroelectrics in this work) triggers the fast diffusion-less mechanism, on which it induces the swift control of polarization and corresponding thermal conductivity through proton-phonon coupling effect. The growth medium of molecular ferroelectrics and the humidity/water content provide a nice pathway towards the versatile tuning and control of ferroelectrics, only and especially for molecular ferroelectrics materials. The authors have carried out comprehensive studies from ferroelectric, thermal and electroresistive perspectives to uncover the novel stimuli tunable molecular ferroelectrics. The ionics using ion-controlled functional materials show a great potential to a conventional electronic platform, while ionics provide an extra ion tunability. One comment for the consideration is to introduce the criteria for ionic tuning (other than proton, any other potential candidates, etc.). Overall, this is a new research direction in molecular ferroelectrics with stimuli tuning ferroelectric/thermal/memory effects using ions. Therefore, I strongly recommend the publication of this article in Nature Communications.

Reviewer #4 (Remarks to the Author):

The authors report stimuli responsive (proton and voltage) molecular ferroelectrics on which its ferroelectricity, thermal transport and electroresistance are tunable via proton ions. The structural, electrical and corresponding ferroelectric measurements provide a unified picture of the operation principles in such stimuli dependent molecular ferroelectrics, resulting from its molecular moieties, aqua environment and large lattice spacing. The ionic molecule ferroelectrics is a novel discovery in this report which could find a broad field potential applications after this (thermal transfer using ferroelectrics, memory, etc.). It would be interesting to provide some additional introductory description/discussion on thermal conductivity tuning using ferroics. Another comment is to provide its potential pathway utilizing thermal tunability.

RESPONSE TO REVIEWERS' COMMENTS

Reviewer #1 (Remarks to the Author):

The authors investigated molecular ionic ferroelectrics exhibiting the coexistence of room-temperature ionic conductivity (6.1×10^{-5} S/cm) and ferroelectricity, which triggers the ionic-coupled ferroelectric properties. Such ionic ferroelectrics with aqua-ligand molecules further present the controlled tunability in polarization from 0.68 to 1.39 C/cm², thermal conductivity by 13 % and electrical resistivity by 86 % due to the proton transfer in an ionic lattice under external stimuli. In addition, the current version of this paper does not provide the insight result for the molecular ferroelectric ACTA PHYSICA SINICA 69 (21)s to guarantee the publication in Nature Communications. The following issue should be addressed before the publication.

Reply: We thank the reviewer for the careful review and comments on our manuscript. We provide a point-by-point response as below.

1) The same material ImClO₄ were reported in the previous paper of these authors. The reference No. 30 is proton conductivity in the paper of Nat. Commun. 12, 255 4602 (2021). However, the same ImClO₄ is also ionic ferroelectrics.

Reply: We thank the reviewer for the comment regarding this manuscript and our previous publication. In our previous work, we found proton switching magnetoelectricity in three-dimensional molecular heterogeneous solids that consist of molecular magnet network and molecular ferroelectric (ImClO₄). In such heterogeneous solid, molecular magnet provides proton reservoir to modulate magnetism, while molecular ferroelectrics can charge proton transfer to reversibly manipulate magnetism. In this manuscript, we report the intrinsic proton in the single-phase molecular ferroelectric ImClO₄, which as the source of ionic conductance provides the access to control ferroelectric properties. therefore, the main finding in this manuscript originates from the intrinsic properties of ferroelectric ImClO₄ crystal, differing from the emerged magnetoelectric coupling in the heterogeneous solid.

2) What is the physics of ionic ferroelectrics? The author should discuss more about the physics.

Reply: Molecular ferroelectrics consist of the displaced cations and anions to induce spontaneous polarization, which are often electric insulator. Ionic ferroelectrics are considered to manifest ionic conductivity together with its ferroelectricity in one single material platform, providing an exciting opportunity to achieve the ionic-controlled ferroelectric behavior (Fig. 1). Aqueous-solution synthesized molecular ferroelectrics possess the water molecule (proton reservoir) networks with hydrogen bonds, on which the proton transfer mediates ionic/proton conductance of molecular ferroelectrics. The proton transfer tailors the dipole and polarization in molecular ferroelectrics (in this case, consisting of Im cation and -ClO₄ anion), which in turn enable the new pathway towards the polarization controlled thermal conductivity and resistive switching in molecular ferroelectrics. In the revision, we add the further discussion on this aspect, and carry out additional experimental results (proton controlled thermal transport and

emission spectra) to support such conclusion (Fig. 4 and Fig. S9 and Table S1).

3) Figures 1 c and d are not London free energy. It could be Landau free energy or total free energy. Normally, the free energy can be represented by F , and can not use E . In addition, the following related references should be cited to describe the free energy vs polarization in molecular ferroelectrics. *Advanced Functional Materials* 31 (38), 2104393, *ACTA PHYSICA SINICA* 69 (21).

Reply: We thank the reviewer for correcting the free energy representation. The symbol has been revised in the manuscript and related references cited for the description.

4) In Figure 3a, the three polarization loops are under different electric field. However, the saturation polarizations are not equal for different electric fields.

Reply: We thank the reviewer for bringing up this point. Polarization varies at different electric fields and reaches the maximum value at a saturated electric field. Those three polarization loops are aimed to show polarization of ImClO_4 changes with electric field, which are not fully saturated. The slightly increased polarization could be attributed to the polarization alignment of ferroelectric domains in ImClO_4 crystal. The polarization in each ferroelectric domain can orient along one direction at a higher electric field. Similar behavior can be found in Fig. S1 in the supporting information as well as the literature report¹, which are plotted in Fig. R1.

Fig. R1. P - E hysteresis loops under different electric fields measured in (a) ImClO_4 crystals cited from Fig. S1 in the supporting information; (b) $\text{Bi}_{0.5}\text{Na}_{0.5}\text{TiO}_3\text{-BaTiO}_3$ ceramics from the literature¹.

5) Could the author explain the difference of phase transition during cooling and heating?

Reply: The phase transition between ferroelectric and paraelectric is characterized by differential scanning calorimetry (DSC) traces, temperature-dependent thermal conductivity and relative permittivity in Fig. 3 of the manuscript. One common feature is the transition temperature determined by each property during cooling is a little lower than that during heating, which is considered as proton effect on thermal properties. According to the response of relative permittivity, the loss of water molecules at above 373.15 K lowers several degrees on the transition temperature. The less proton sources result in relatively large change in thermal conductance due to electron/proton-phonon coupling.

Reviewer #2 (Remarks to the Author):

Huang *et al.* reported the molecular ionic ferroelectric ImClO_4 , which exhibits coexistence of ionic conductivity and ferroelectricity. Interestingly, the coexistence of ionic conductivity and ferroelectricity can trigger the ionic-coupled ferroelectric properties, *i.e.*, ionic controlled ferroelectric behavior. Because ferroelectric-ion coupling is of great significance in the development of the devices with multi-well polarized states and neuromorphic characteristics, the starting point of this work is interesting and worthy of affirmation. However, there are still many problems to be clarified or solved in the manuscript before publication of the work. The main problems are as follows:

1) The polarization value of the P-E curve in the manuscript is much lower than that of the same compound reported in the literature, and even lower than that of the same compound reported by the authors themselves (*Sci. Adv.* 2017, 3, e1701008), why?

Reply: We thank the reviewer for the careful review. The polarization loops in this manuscript were measured on ImClO_4 crystals, while our previous results were obtained on thin film samples. Our measured polarization in ImClO_4 crystal is about $2.4 \mu\text{C}/\text{cm}^2$ at room temperature, which is a little lower than the reported $\sim 7.5 \mu\text{C}/\text{cm}^2$ in the thin film sample (*Sci. Adv.* 2017, 3, e1701008). The defects and domain boundaries in bulk crystal samples play an important role in its polarization value (including the literature, like $1.1 \mu\text{C}/\text{cm}^2$ by Pajak, Z.'s work²). Therefore, the measured polarization value is related to sample form and its synthetic methods.

2) The change of P-E curve with humidity shows that with the increase of humidity, P-E curve appears certain leakage. Is it due to other contributions, rather than an actual increase in polarization?

Reply: We thank the reviewer for this comment. To address this comment, we provide an additional experiment by using the PUND (positive up negative down) measurement (Fig. R2), which indicates the enhanced polarization due to the humidification process. The humidification enhances dielectric constant of ImClO_4 at the entire frequency range from 20 Hz to 2 MHz (Fig. S3).

Fig. R2. PUND measurement on ImClO_4 crystal at different humidity.

Fig. R3. Frequency dependent dielectric constant under different humidity.

3) *Under humidity control, does water adsorb on the surface or enter the lattice? How to exclude the effect of water on simple surface adsorption of the sample?*

Reply: Water molecules adsorb on the surface which tailors the proton transfer in the crystal. The proton transfer is the bulk effect in the ImClO₄ crystal, where its ionic, polarization and thermal properties are modified by proton (water molecules). Surface adsorption of the water molecules play an important role in proton manipulation, while maintaining the crystal structure of molecular ferroelectrics. As a follow-up on this comment, we have carried out the additional XRD studies (Fig. 4d) and associated discussion in the revised main text by illustrating the crystal structure integrity under the humidification process.

4) *Does ImClO₄ sample itself absorb moisture? Will the surface of the sample be damaged during this process?*

Reply: The ImClO₄ crystal absorbs moisture to stimulate the proton effect controlled by humidity. When we conducted the humidity-controlled experiments, humidity was controlled in a limited level to protect sample from damage. As shown in Fig. R4, X-ray diffraction (XRD) reveals the same crystal structure in pristine and humidified ImClO₄ crystals. The optical images of ImClO₄ crystal (the insets in Fig. R4) don't show change before and after humidity control, except the surface gloss due to the surface absorption. The XRD patterns are overlapped without any shifts. The subtle change of (101) peak intensity could be attributed to the water molecules on the surface. Moreover, Raman spectroscopy, as a surface analysis technique, has not detected changes on vibrational modes in ImClO₄ crystal under different humidity (Fig. R5). Therefore, the absorbed moisture does not damage the surface of ImClO₄ crystal according to the XRD patterns and Raman spectra.

Fig. R4. X-ray diffraction patterns of ImClO₄ crystal before and after humidification. (a) XRD patterns maintain the same structure in the pristine (17%RH) and humidified (70%RH) ImClO₄ crystals. The inset optical images show the surface without damage due to moisture. (b) Enlarged figure indicates that XRD diffraction patterns are almost overlapped, except the (101) peak at low angle.

Fig. R5 Raman spectra of ImClO₄ crystal under different relative humidity. (a) Raman spectra were measured from 3200 cm^{-1} to 100 cm^{-1} . The relative humidity varies from 25 %, 35 %, 40 %, to 70 %. (b) Enlarged Raman spectra at different Raman shift ranges indicate no obvious changes by humidity.

5) *In what way is the thermal conductivity measured? As we all know, it is difficult to obtain accurate thermal conductivity data, please provide details of this part, especially the thermal conductivity test before and after polarization and at different humidity.*

Reply: We thank the reviewer for this comment. The thermal conductivity was measured on a hot disk TPS instrument. Since the ImClO₄ sample is spontaneously polarized at room temperature, the thermal conductivity measured during heating was starting from a polarized state. When cooling from high temperature, the sample was in a unpolarized state. In the Fig. 3c of the manuscript, it can be certain that the thermal conductivity during cooling gets smaller than that during heating and the anomaly shifts. As for the thermal conductivity measured at different humidity, we conducted those measurements at room temperature and kept samples in the controlled humidity levels during measurement (Fig. 4c in this manuscript). Those measurement were repeated at

both low and high humidity to confirm the reliability. The fluctuation of measured thermal conductivity at each humidity is kept in a subtly small range, allowing to distinguish the thermal conductivity at low and high humidity. We also added this description into the revised manuscript.

In addition, the thermal conductivity measurements have been characterized and calibrated extensively for the error analysis. Two techniques, including heat flow meter (ASTM C518) and hot disk TPS (ISO standard 22007-2), are utilized for the calibration using the NIST-standard polystyrene polymer reference material (with certificate, including NIST SRM 1450d). The TPS technique, now commercially available, serves as a non-destructive method for the measurement of thermal conductivity in a wide range ($0.010 \text{ W m}^{-1} \text{ K}^{-1} \sim 500 \text{ W m}^{-1} \text{ K}^{-1}$) for a variety of sample forms, such as thin film specimens and anisotropic materials. It also does not need metal sensors or transducer layer coatings on the membrane sample which is necessary in traditional $3-\omega$, or optical-based methods. Error in the TPS measurement can potentially come from two sources: (1) uncertainty in the experimental data and the selection of time interval for analysis, and (2) deviation of the original idealized analytical heat transfer model from the practical measurement scenario. We chose the appropriate time interval to calculate thermal conductivity and simultaneously make sure all other parameters (BG thermal conductivity, BG thermal diffusivity, BG specific heat, probing depth, temperature increase, temperature drift, total to characteristic time, total temperature increase, time correction, mean deviation, and sensor resistance) within a range. 2) The standard NIST-standard polystyrene polymer reference material (with certificate, including NIST SRM 1450d) have been applied as the control sample to calibrate and verify the accuracy of TPS instrument. All standard samples have also been confirmed and calibrated by using the steady-state measurements to verify its thermal conductivity from the certified values. The thermal conductivity results obtained in this work agree well with other measurement techniques and international standard polymeric materials. This agreement indicates that the method we applied in this work is accurate to within the error bar (within $\pm 5\%$) over a thermal conductivity range. The error analysis was carried out by testing multiple samples and several times to produce the resulting reported average.

6) In Figure 3, the author attributes the dielectric anomaly in a very wide temperature region near 300 K to the mobile proton released from the framework. However, this anomaly also exists in the heating stage of DSC, which is attributed to the structural phase transition. Why?

Reply: As we observed in the temperature dependent dielectric constant, the wide anomaly occurs near 300 K which is in the middle region of structural transition (~ 220 K) and ferroelectric transition (372 K). The dielectric anomaly near 300 K is attributed to the released proton ions during heating. In DSC measurement, the heating curve seems flat near 300 K. One big anomaly occurred at the ferroelectric transition near 365 K, when sample was cooled. The structural transition occurs at 212 K (cooling) and 246 K (heating) from DSC measurement that is consistent with the literature².

7) In Figure 5b, the P - E loops are enhanced by relatively decreasing resistivity. However, the change is too small to be convincing.

Reply: The change of P - E loops by stimuli control is indeed small, where the polarization results from the displacement of cations and anions in the molecular crystal lattice. Electric stimuli slightly modify the cations and anions, as well as protons in the lattice, which induce a small change on polarization. Besides, this polarization change is reversible by stimuli, suggesting the stimuli effect (Fig. R6). The P - E loop also recovers into the initial shape (Fig. R6), which also has been added in the Fig. S12d in the supporting information. We also address this in the revised manuscript.

Fig. R6. Electric current stimuli effect on P - E loops of ImClO_4 crystal. A decreased resistivity (from 1040 to 504 $\text{M}\Omega\cdot\text{cm}$) results in an enhanced P - E loop. ImClO_4 crystal recovered into a high resistivity (1200 $\text{M}\Omega\cdot\text{cm}$), meanwhile corresponding P - E loop shrunk back to the initial shape.

8) For Figures S5 and S6, the authors claim that “The repeatable permittivity peak at ferroelectric ordering temperature indicates high crystalline quality of as-grown ImClO_4 , further evidencing the proton mechanism for the anomaly in permittivity which relates to its cooling history and magnetic field dependence (Figs. S5 and S6).” However, the authors have been discussing dielectric anomalies around 300 K. Because the phase transition temperature is about 373 K, we have not yet seen repeatable peaks in permittivity at ferroelectric ordered temperatures. How to explain the variation of dielectric constant under cooling history and magnetic field dependence by proton mechanism? The author needs to provide a detailed explanation.

Reply: We thank the reviewer for this comment and point out this error. We update the main text accordingly, on which the repeatable permittivity peak is the one near 300 K, not the ferroelectric transition peak. We update this sentence in the revised manuscript, “The permittivity anomaly near 300 K is repeatable and dependent on magnetic field, further suggesting the proton mechanism which relates to its cooling history (Figs. S5 and S6)”. The anomaly peak can shift according to the lowest temperature where ImClO_4 crystal was heated up (Fig. S5), as well as varies in magnitude under different magnetic field (Fig. S6).

Reviewer #3 (Remarks to the Author):

This work reports the ionic tuning of molecular ferroelectricity, thermal transfer and electronic memory, which is a new exciting frontier for molecular ferroelectrics. The proton transfer in molecular crystals (ferroelectrics in this work) triggers the fast diffusion-less mechanism, on which it induces the swift control of polarization and corresponding thermal conductivity through proton-phonon coupling effect. The growth medium of molecular ferroelectrics and the humidity/water content provide a nice pathway towards the versatile tuning and control of ferroelectrics, only and especially for molecular ferroelectrics materials. The authors have carried out comprehensive studies from ferroelectric, thermal and electroresistive perspectives to uncover the novel stimuli tunable molecular ferroelectrics. The ionics using ion-controlled functional materials show a great potential to a conventional electronic platform, while ionics provide an extra ion tunability. One comment for the consideration is to introduce the criteria for ionic tuning (other than proton, any other potential candidates, etc.). Overall, this is a new research direction in molecular ferroelectrics with stimuli tuning ferroelectric/thermal/memory effects using ions. Therefore, I strongly recommend the publication of this article in Nature Communications.

Reply: We thank the reviewer for the careful review and recommendation.

Reviewer #4 (Remarks to the Author):

The authors report stimuli responsive (proton and voltage) molecular ferroelectrics on which its ferroelectricity, thermal transport and electroresistance are tunable via proton ions. The structural, electrical and corresponding ferroelectric measurements provide a unified picture of the operation principles in such stimuli dependent molecular ferroelectrics, resulting from its molecular moieties, aqua environment and large lattice spacing. The ionic molecule ferroelectrics is a novel discovery in this report which could find a broad field potential applications after this (thermal transfer using ferroelectrics, memory, etc.). It would be interesting to provide some additional introductory description/discussion on thermal conductivity tuning using ferroics. Another comment is to provide its potential pathway utilizing thermal tunability.

Reply: We thank the reviewer for the comments. We added some discussion after Fig. 4c in the revised manuscript, considering the thermal conductivity tunability by electric field. Besides, this proton-controlled thermal conductivity provides the tunability in a non-contact mode, which is beneficial for precise devices in the field of thermal circuit and thermal management. This discussion has also been added in the revised manuscript. Thanks for reviewing our work again.

References

1. Shi, J., Fan, H., Liu, X. & Li, Q. Ferroelectric hysteresis loop scaling and electric-field-induced strain of $\text{Bi}_{0.5}\text{Na}_{0.5}\text{TiO}_3\text{-BaTiO}_3$ ceramics. *physica status solidi (a)* **211**, 2388-2393 (2014).
2. Pajak, Z., Czarnecki, P., Szafranska, B., Maluszynska, H. & Fojud, Z. Ferroelectric ordering in imidazolium perchlorate. *J Chem Phys* **124**, 144502 (2006).

REVIEWERS' COMMENTS

Reviewer #1 (Remarks to the Author):

The revised manuscript can be accepted as it is.

Reviewer #2 (Remarks to the Author):

The author's response to my concerns was handled with seriousness. I am satisfied with the author's revised manuscript and recommend the publication of this article in Nature Communications.

Reviewer #3 (Remarks to the Author):

The raised points have been well solved.

Reviewer #4 (Remarks to the Author):

The authors already revised according to the comments of reviewers and the current manuscript can be accepted for formal publication.

RESPONSE TO REVIEWERS' COMMENTS

Reviewer #1 (Remarks to the Author):

The revised manuscript can be accepted as it is.

Reply: We thank the reviewer again for reviewing our manuscript and recommending its publication.

Reviewer #2 (Remarks to the Author):

The author's response to my concerns was handled with seriousness. I am satisfied with the author's revised manuscript and recommend the publication of this article in Nature Communications.

Reply: We thank the reviewer for all the previous comments and the recognition of our work that allow the publication.

Reviewer #3 (Remarks to the Author):

The raised points have been well solved.

Reply: We thank the reviewer for the recommendation.

Reviewer #4 (Remarks to the Author):

The authors already revised according to the comments of reviewers and the current manuscript can be accepted for formal publication.

Reply: We thank the reviewer for the careful review again and the recommendation for publication.